# Molecular Traits and Functional Exploration of *BES1* Gene Family in Plants

**DOI:** 10.3390/ijms23084242

**Published:** 2022-04-11

**Authors:** Zhenting Sun, Xingzhou Liu, Weidong Zhu, Huan Lin, Xiugui Chen, Yan Li, Wuwei Ye, Zujun Yin

**Affiliations:** 1Zhengzhou Research Base, State Key Laboratory of Cotton Biology, School of Agricultural Sciences, Zhengzhou University, Zhengzhou 450001, China; 13283882072@163.com (Z.S.); chenxiugui@caas.cn (X.C.); 2Suzhou Academy of Agricultural Science, Suzhou 234000, China; lxz8610@163.com; 3Shenzhen Branch, Guangdong Laboratory for Lingnan Modern Agriculture, Genome Analysis Laboratory of the Ministry of Agriculture, Agricultural Genomics Institute at Shenzhen, Chinese Academy of Agricultural Sciences, Shenzhen 518120, China; yuanze1003@163.com; 4State Key Laboratory of Cotton Biology, Institute of Cotton Research, Chinese Academy of Agricultural Sciences, Anyang 455000, China; 82101195031@caas.cn (H.L.); hai-19@163.com (Y.L.)

**Keywords:** *BES1* genes, rice, cotton, transgenic plants, collinearity analysis, abiotic stress

## Abstract

The *BES1* (BRI1-EMSSUPPRESSOR1) gene family is a unique class of transcription factors that play dynamic roles in the Brassinosteroids (BRs) signaling pathway. The published genome sequences of a large number of plants provide an opportunity to identify and perform a comprehensive functional study on the *BES1* gene family for their potential roles in developmental processes and stress responses. A total of 135 *BES1* genes in 27 plant species were recognized and characterized, which were divided into five well-conserved subfamilies. *BES1* was not found in lower plants, such as *Cyanophora paradoxa* and *Galdieria sulphuraria*. The spatial expression profiles of *BES1s* in Arabidopsis, rice, and cotton, as well as their response to abiotic stresses, were analyzed. The overexpression of two rice *BES1* genes, i.e., *OsBES1-3* and *OsBES1-5*, promotes root growth under drought stress. The overexpression of *GhBES1-4* from cotton enhanced the salt tolerance in Arabidopsis. Five protein interaction networks were constructed and numerous genes co-expressed with *GhBES1-4* were characterized in transgenic Arabidopsis. *BES1* may have evolved in the ancestors of the first land plants following its divergence from algae. Our results lay the foundation for understanding the complex mechanisms of *BES1*-mediated developmental processes and abiotic stress tolerance.

## 1. Introduction

Plant development is synergistically controlled using various gene complex networks. The steroid hormones brassinosteroids (BRs) play important roles in regulating diverse plant processes such as cell elongation, photomorphogenesis, and reproduction, as well as both abiotic and biotic stress responses. BRI1-EMSSUPPRESSOR1 (BES1), as a new plant-specific transcription factor (TF), plays a significant role in modulating BR-regulated gene expression [1,2]. The *BES1* protein comprises three domains including the BRASSINOSTEROID INSENSITIVE 2 (BIN2) phosphorylation domain (P), amino-terminal domain (N), and carboxyl-terminal domain [3]. It was reported that the N-terminal domain comprises a conserved motif and a nuclear localized sequence that combine and assist as a DNA binding structure to unite with the basic helix-loop-helix (bHLH) domains of BES1-interacting Myc-like 1 (BIM1). The P domain is well known as a target of the BIN2 kinase, but little is known about its biological functions [4].

Regulation of *BES1* occurs in multiple routes, among which the phosphorylation of *BES1* proteins is well studied. BIN2 acts as an inhibitor of BR signaling through interaction with *BES1* to regulate BR signal transduction via its phosphorylation and dephosphorylation in Arabidopsis (*Arabidopsis thaliana*) [5]. When BRs are at low levels, BIN2 phosphorylates *BES1* and prevents its nuclear localization, suppresses its DNA binding activity, and/or promotes its degradation [6]. *BES1* and BRASSINAZOLE RESISTANT1 (BZR1) contain two conserved lysine residues, K280 and K320, which serve as the conjugation sites of small ubiquitin-like modifier (SUMO), which modifies *BES1* post-translationally and alters its functionality [7]. A previous study indicated that *BES1* is a major factor that contributes either transcriptional activation or repression to several target genes in Arabidopsis. *BES1*, coupled with BZR1, binds to the promoter of AGAMOUS-LIKE15 (AGL15), directly repressing its transcription, thereby maintaining its low level of expression that is necessary in the seed maturation program during the seed-to-seedling transition [8]. It also repressed the expression of CIRCADIAN CLOCK-ASSOCIATED 1 (CCA1) and LATE ELONGATED HYPOCOTYL (LHY) in the dark phase of the diurnal cycle by binding to their promoters, through its interaction with TOPLESS (TPL) [9]. In apple, *BES1* positively regulated the expression of *MYB88* under pathogen attack. The downregulation of *MdMYB88* in the plants overexpressing *MdBES1* decreased the resistance to a pathogen and C-REPEAT BINDING FACTOR1 expression [10]. *BES1* regulates a number of genes involved in various physiological processes. The sensitivity of the *AtBES1* mutant was related to the reduction in photochemical efficiency of PSII and increased the content of tocopherol and lipid hydrogen peroxide, namely, the increased photoinhibition and photooxidation stress during heat stress. *BES1* can interact with ABSCISIC ACID INSENSITIVE5 (ABI5) and significantly downregulates the expression of the downstream regulatory genes by inhibiting the binding of ABI5 to the promoter regions of these genes, which promotes seed germination [11]. In maize, *BES1/BZR1-5* positively regulates kernel size. The overexpression of *ZmBES1*/*BZR1-5* significantly increased the seed size and weight in maize, which was verified in the Mu transposon insertion mutant that produces smaller kernels [12]. TINY (encodes a member of the DREB subfamily A-4 of ERF/AP2 transcription factor family), an APETALA2/ETHYLENE RESPONSIVE FACTOR TF that is inducible by drought stress, antagonizes *BES1,* leading to the inhibition of BR-regulated growth and the enhancement of drought-responsive gene expression [13]. Strigolactones (SLs) are a class of terpenoid phytohormones, which has been found to regulate shoot branching in both rice and Arabidopsis, in addition to the aforementioned roles in BR signaling [14].

With the rapid progress in whole genome sequencing, which has been completed in a great number of plants, *BES1* genes have been profiled in numerous plant species [15,16,17]. However, the advancement of the comprehensive understanding of the *BES1* family genes in the plant kingdom is clearly lacking. This study aimed to carry out a comprehensive investigation of the *BES1* gene family in algae, bryophytes, lycophytes, and vascular plants. The distribution, phylogenetic tree, gene structure, and collinearity of the *BES1* gene family were analyzed to understand the evolutionary history of *BES1* in plants. The spatial gene expression patterns of *BES1*, together with their responses to various abiotic stresses, were also analyzed in Arabidopsis, rice, and cotton. Furthermore, co-expression and transgenic research were examined in some *BES1* members. The outcome of this study may shed more light on the functionality and functional networks of the *BES1* gene family and lay a solid foundation for the future exploration of its potential applications in biological research and agricultural production.

## 2. Results

### 2.1. Identification of BES1 Gene Family in Plantae

The examination and characterization of the *BES1* gene family in 27 species in the Plantae Kingdom, signifying nine main plant families including rhodophytes, chlorophytes, glaucophytes, bryophytes, gymnosperms, lycophytes, basal angiosperms, eudicots, and monocots, were chosen for this investigation (Table 1). A total number of 135 *BES1* genes, following filtering, were identified, and the breakdown of the numbers per specie is shown in Table 1. No *BES1* gene was identified in the *Cyanophora paradoxa*, *Galdieria sulphuraria*, *Cyanidioschyzon merolae,* and 7 Chlorophytes species. The recognized *BES1* genes presented deviated molecular features with protein lengths ranging from 158 to 801 amino acid residues. They are randomly distributed on the chromosomes in each species (Appendix A). There is no correlation between the number of *BES1* genes and genome size of the plant species. For example, there are 4 *BES1* genes in *Selaginella moellendorffii* that has a genome size of 100 Mb, whereas there are a similar number of *BES1* genes in *Picea abies* that has a genome size of 19,600 Mb. As a tetraploid, *G**. hirsutum* (*Gossypium hirsutum*) contains the most *BES1* genes (22), being the sum of the number of *BES1* genes identified in its two diploid progenitors, i.e., *G**. arboreum* (*Gossypium arboreum*) and *G**. raimondii* (*Gossypium raimondii*) (11).

### 2.2. Phylogenetic Analysis, Conserved Motif, and Protein Characteristics

To uncover whether and when natural selection had acted on the evolution of the *BES1* gene family, we aligned the 135 identified *BES1* protein sequences and constructed a phylogenetic tree. These genes were grouped into five clades (Figure 1A). All the *BES1s* in the Bryophytes and Lycophytes were assigned into Sub II and III. None of the angiosperm *BES1s* were detected in Sub III, suggesting that the genes of Sub III might have experienced an evolutionary divergence in structure or function. Sub I, IV, and V contained 33, 30, and 37 proteins, respectively, which were mainly from the higher terrestrial plant species belonging to monocotyledonary and dicotyledonary plants. There was only one *BES1* in the basal angiosperm (*Amborella trichopoda*) in Sub I and IV, and two *BES1* in Gymnosperms (*Picea sitchensis*) in Sub V.

The ratio of Ka/Ks can be used to determine whether there is a selective pressure acting on this protein-coding gene. In *P. patens*, *M. polymorpha*, *S. moellendorffii*, *A*. *trichopoda*, Arabidopsis, and three cotton species, the Ka/Ks values were calculated for 80 gene pairs (Figure 1B), but none in *P. patens*, *M. polymorpha,* and *S. moellendorffii*. Among other species, the Ka/Ks values of 75 gene pairs were less than 0.5, whereas those of four gene pairs were between 0.5 and 1.0, suggesting that strong purifying selection pressure might have occurred in these species. There was one gene pair that showed a Ka/Ks value greater than 1.0, suggesting directional selection.

*BES1* contains a highly basic region that is very similar to the basic regions of other bHLH proteins. The Glu-13Arg-16 pair and two leucine residues are also present in all the *BES1* family members. Although every protein can form a helix-loop-helix structure as predicted, variation exists in five subgroups (Figure 1C). We selected one *BES1* gene in each subgroup to understand the physicochemical properties, among which *GhBES1-19*, XP_024375445.1, ADE77805.1, *GhBES1-4,* and *GhBES1-1* were selected from Sub I to Sub V. All these proteins possess hydrophilic features. The disordered structure of *GhBES1-19* and XP_024375445.1 accounted for a large proportion (about 40%), while the disordered structure of ADE77805.1, *GhBES1-4,* and *GhBES1-1* accounted for 74–79%. The Alpha helix of *GhBES1-19*, XP_024375445.1, ADE77805.1, *GhBES1-4,* and *GhBES1-1* was 36%, 38%, 14%, 13%, and 18%, respectively (Appendix A).

### 2.3. Structure, Collinearity, and Cis-Acting Elements Analysis

The exon/intron structure of paralogous genes is highly conserved, which could be used to elucidate the evolutionary relationships between species [4]. The variation and diversity in the *BES1* gene structural features in plant species and their underlying molecular mechanism are summarized in Appendix A. Interestingly, *BES1* family participants shared a parallel gene structure even within the same subfamily according to the number of intron and intron phases except Sub I within which four family members have only one exon, six members have two exons, and the other members have nine exons. This is in contrast to the most members in Sub II, III, IV, and V, which contain two exons.

The collinearity analysis of *BES1* genes in Arabidopsis, rice, and cotton showed that seven *AtBES1* genes were correlated with twelve *GhBES1* genes, and three *OsBES1* genes were correlated with three *GhBES1* genes (Figure 2). Among them, six *AtBES1* and two *OsBES1* genes were related to at least two cotton *BES1* genes. The collinearity between the three cotton species (*G. arboretum*, *G. raimondii,* and *G. hirsutum*) was also established (Figure 2), showing that *G. hirsutum* formed 23 and 35 homologous gene pairs with *G. arboretum* and *G. raimondii*, respectively. *GhBES1* genes had a collinear relationship with multiple *GaBES1* and *GrBES1* genes, indicating that the increase in number of *GhBES1* gene was not only due to the increase in genome size, but also the splicing and replication that have occurred between chromosomes.

*Cis*-acting elements play an important role in the regulation of gene transcription initiation. In this study, all of the *BES1* genes showed a light response. A total of 17 genes are relevant to the production of salicylic acid. In addition, numerous elements in response to abscisic acid, auxin, defense and stress, gibberellin, and MeJA have been recognized, implying their regulation in hormones or stress responses (Appendix A).

### 2.4. Expression Analysis of BES1 Family Genes in Arabidopsis, Rice, and Cotton

To detect the functional divergence, the spatial and temporal expression patterns of the *BES1* genes were investigated in Arabidopsis, rice, and cotton. In Figure 3, the transcriptomic levels of *AtBES1*, *OsBES1,* and *GhBES1* genes were evaluated in both the vegetative and reproductive tissues, which showed clear tissue-specificities. In Arabidopsis, *AtBES1-7* showed the highest expression level in the largest number of tissues and developmental phases of plants than its paralogues. *AtBES1-2* and *AtBES1-8* were mainly expressed in the seedlings and young rosettes, whereas *AtBES1-3* was significantly articulated in the young rosette. The rest of the *AtBES1* family genes showed relatively low expression levels in most tissues/developmental stages. In rice, *OsBES1-5*, *OsBBES1-4*, *OsBES1-2*, and *OsBES1-6* transcripts showed higher expression levels compared to *OsBES1-1* in almost all plant tissues and developmental phases, despite the variation in expression patterns. The variation in expression of *OsBES1-3* was only discernible at the reproductive stages.

Among the cotton *GhBES1* genes, *GhBES1-4* and *GhBES1-12* were highly expressed in all plant tissues, in contrast to *GhBES1-3* and *GhBES1-11* that were not detected in any of the examined tissues. The expression of *GhBES1-6*, *-7*, *-13*, *-14*, *-16,* and *-22* was barely discernible during fiber development. It is known that the transcript abundance of a gene in a specific plant tissue establishes the important clues for its biological functions. For instance, *GhBES1-4* and *GhBES1-12* were gene paralogues dominantly expressed during ovule development, advising that they may share a conserved functional role in seed development. 

### 2.5. Expression Profiles of BES1 Family Genes under Abiotic Stresses

The mRNA levels of *AtBES1*, *OsBES1*, and *GhBES1* genes were induced by abiotic stresses. In brief, cold and heat stresses significantly up-regulated the transcription levels of *AtBES1-2*, *AtBES1-1*, *AtBES1-3,* and *AtBES1-4*, but showed nonsignificant effects on the expression of *AtBES1-5*, *AtBES1-7*, *AtBES1-6*, *AtBES1-8*. PEG treatment markedly up-regulated the transcription of all the *AtBES1* genes, especially *AtBES1-6* and *AtBES1-8*. NaCl treatment greatly up-regulated the transcriptions of *AtBES1-2* and *AtBES1-1*, and down-regulated those of *AtBES1-5*, *AtBES1-7*, *AtBES1-6*, and *AtBES1-8*, without discernible effects on the expressions of *AtBES1-3* and *AtBES1-4* (Figure 4A).

The expression patterns of *OsBES1* family gene members displayed a clear temporal and spatial diversification (Figure 4B). Under cold and PEG stresses, almost all the *OsBES1* genes were significantly up-regulated in both the roots and shoots following a period of 12 h of treatment, except *OsBES1-4* that showed relatively moderate up-regulation in response to these stresses. Under 12 h of heat treatment, *OsBES1-6*, *OsBES1-5*, *OsBES1-3*, and *OsBES1-2* were up-regulated in shoots but the expressions of *OsBES1-4* and *OsBES1-1* were not affected. In roots, all the *OsBES1* genes maintained moderate changes at 12 h of heat treatment. Under 12 h of NaCl treatment, significantly higher levels of mRNA transcripts were observed from all of the *OsBES1* family genes in shoots, but not in roots. It should be noticed that the expression patterns of both *AtBES1* and *OsBES1* genes did no show distinct subfamily characteristics under the treatment of these environmental stresses.

Figure 4C represents the heat map analysis of the *GhBES1* family genes’ responses to the environmental stresses such as cold, hot, dehydration, and salinity. *GhBES1-3* and *-11* were barely discernible under all the four stress treatments. Under low-temperature treatment, *GhBES1-1*, *-2*, *-4*, *-5*, *-7*, *-8*, *-9*, *-10*, *-12*, *-13*, *-15*, and *-17* showed an initial low expression before raising to high levels, which is opposite to the trend of responsiveness to cold treatment by *GhBES1-6*, *-14*, and *-16*. Under high-temperature treatment, *GhBES1-1*, *-2*, *-5*, *-7*, *-8*, *-9*, *-10*, *-13*, and *-16* displayed an initial increase before a reduction in expression, but *GhBES1-4*, *-12*, *-14*, *-15*, *-17*, *-18*, *-19*, *-20,* and *-21* showed a constant reduction trend. Under PEG treatment, the expressions of *GhBES1-1*, *-2*, *-8*, *-9*, *-13,* and *-19* were increased, but the opposite was true to *GhBES1-4*, *-10*, *-14*, *-15,* and *-16*. In response to salinity treatment, the expressions of *GhBES1-1*, *-2*, *-5*, *-7*, *-8*, *-9*, *-10*, *-18,* and *-19* showed a trend of increase, but *GhBES1-4*, *-6*, *-14*, *-15*, *-16*, *-17,* and *-22* showed a trend of reduction. Furthermore, the expression of *GhBES1-1* and *-4* was rapidly increased to very high levels in response to different stress treatments, suggesting their potentially crucial roles in managing abiotic stresses. Nine *GhBES1* genes were selected for qRT-PCR analysis, which showed consistent expression patterns and verified the transcriptome analysis (Appendix A).

### 2.6. Functional Verification of Transgenic Plants 

Both *OsBES1-3* and *OsBES1-5* showed the response against drought both in shoots and roots. To further examine their functions, transgenic rice overexpressing these two genes (*pUBI::OsBES1-3* and *pUBI::OsBES1-5*) were generated for the stress tolerance test. To determine the gene’s effects on osmotic tolerance, the seeds of the transgenic rice overexpressing *OsBES1-3* or *OsBES1-5* rice, together with the WT control, were germinated on plates, followed by treatments with 20% PEG6000. All the transgenic plants were able to overcome the inhibition of PEG6000 to some extent, and showed significantly longer root lengths than WT plants did (Figure 5A,B). The seedlings were cultured in plastic pots for 4 weeks and then treated with 20% PEG6000. After the treatment for one week, most of the WT seedlings were wilting and yellow (Figure 5D), and the survival rate of the WT plants was only 54.3%, whereas the transgenic rice plants remained green and survived (Figure 5C).

To investigate *GhBES1′s* potential functional role in NaCl stress tolerance, *GhBES1-4* was chosen to be overexpressed due to its relatively high expression level in developmental stages and during the abiotic stresses. When the seeds were treated with 175 mM NaCl for about ten days, transgenic *GhBES1-4* Arabidopsis could grow and form roots normally, in sharp contrast to WT that barely grew with only a few seedlings having short roots (Figure 6A). Under salt stress, WT and *GhBES1-4(OE)* showed a significant difference in survival rate (Figure 6B). As shown in Figure 6C, the fully grown transgenic plants also displayed considerable resistance to salt stress relative to WT.

*GhBES1-4* was also transiently expressed in *N. benthamiana* (*Nicotiana benthamiana*) leaf, which displayed a higher resistance to salt stress than the empty vector with the control (Appendix A). Following the treatment with 250 mM NaCl for about ten days, the leaves of transiently expressing *GhBES1-4* maintained a significantly higher level of chlorophyll (88.5%) as compared to the control (Appendix A). In contrast, the control leaves contained less chlorophyll under the salinity treatment, with 58.7% of its water control.

### 2.7. Construction of Protein Interaction Network and Gene Co-Expression

Investigation of the protein interaction networks is an effective way to understand protein interactions and regulatory relationships (Figure 7A–E). Four BES1 proteins in Arabidopsis (AtBES1-2, AtBES1-1, AtBES1-3, and AtBES1-8) and one BES1 protein in rice (OsBES1-3) were randomly selected to predict their interaction network. These BES1 proteins corresponded to the five BES1s in *G. hirsutum* (GhBES1-4, GhBES1-9, GhBES1-12, GhBES1-14, and GhBES1-20), all of which were found to interact with ten other proteins, as exemplified by the AtBES1-2 and AtBES1-3 that were interacting with DWF4, BZR1, BIN2, and BRI1.

In cotton leaves, GhBES1-4 exhibited a significant up-regulation under salt stress based on iTRAQ data. The absolute fold-change of ’Salt/CK’ was 1.25. The value of ’Ratio Salt/CK’ of the three replicates was 1.180|1.464|1.103. The expression patterns of the ten interactive genes were analyzed in transgenic Arabidopsis overexpressing *GhBES1-4* under the treatment with 175 mM NaCl. A total of eight genes including *IWS1* (Interacts with SUPT6H 1), *ELF6* (Early Flowering 6), *BIM1*, *MYB30*, *BZR1*, *BIN2*, *BRI1,* and *BKI1* showed up-regulated expressions in transgenic plants relative to WT plants, whereas the other two genes, i.e., *BSU1* and *DWF4,* were down-regulated (Figure 7F). 

## 3. Discussion

The plant-specific transcription factors *BES1* displayed key roles in the BR signaling network, but a systematic study in plantae is clearly lacking [18]. Overall, 135 *BES1* genes were obtained from the representing nine major plant lineages (Table 1), while it was not identified in the *Cyanophora paradoxa*, *Galdieria sulphuraria*, *Cyanidioschyzon merolae,* and 7 Chlorophytes species. Amborellales and Nymphaeales belong to the so-called ANA-grade of angiosperms, which are extant representatives of lineages that diverged the earliest from the lineage leading to the extant angiosperms. Both *A. trichopoda* and *N. colorata* have nine *BES1* genes, without a difference in the average number of *BES1* genes from monocots and eudicots. These results suggested that there was no spread presence of *BES1* genes within the early-diverging flowering plants than previously anticipated. The variation in the average number of *BES1* genes among the nine main plant families manifests that the genetic mechanism underlying the evolution of these genes was distinct in each group. Although the number of *BES1* genes in *G. hirsutum* was twice as many as in its two diploid progenitors, *G. arboreum* and *G. raimondii*, not every *GaBES1* and *GrBES1* is associated with the *GhBES1* gene. This indicated that the number of *GhBES1* genes does not simply reflect the expansion with their shared genome duplication event (Figure 2).

The *BES1* gene family could be separated into five groups numbered from Sub I to V in accordance with the topology and the deep duplication nodes (Figure 1). Sub II and III contain all BES1 proteins from Bryophytes and Lycophytes, indicating that these two subfamilies, especially Sub II, are the oldest of the five subfamilies. The *BES1* family members from the higher plant species were clustered into other subfamilies, demonstrating that the *BES1* family initiated after the separation of algae and the ancestors of land plants. *BES1* genes from the same lineage inclined to be grouped together in the phylogenetic tree, suggesting their common ancestry and duplications after speciation. Genetic structure analysis showed that most of the members have an intron except Sub I. In addition, tandem repeat events cause the increase in the number of introns and the generation of new genes [19]. Sub I has more introns compared to others, suggesting functional divergence. 

The upstream sequences of the *BES1* gene harbors a variety of *cis*-acting elements (Appendix A), including stress, hormone, and light response elements, indicating its potential involvements in a variety of stress and hormone response processes as a mechanism in promoting plant growth and development and stress tolerance. During the plant life cycle or in different plant tissues, the expressions of *BES1* genes show different patterns even in the same subfamilies. In Sub V, the *BES1* genes in rice (*OsBES1-2* and *OsBES1-4*) showed consistent expression throughout the plant phases. In contrast, *AtBES1* genes in Sub V were only abundantly expressed at certain stages (Figure 3A,B). In Sub IV, *GhBES1-4* and *GhBES1-12* were constitutively expressed at very high levels in all tissues tested, indicating the significant regulatory roles they may play during multiple developmental stages. *GhBES1-2* and *-12*, also belonging to Sub IV, were highly expressed only in the ovule and fiber (Figure 3C). A number of recent studies have demonstrated that the *BES1* TFs regulate plant architecture, root development, and promotion of cell elongation [20,21]. Here, the *OsBES1-1*, *-2*, *-3,* and *-4* showed increased expression in the shoot, but decreased expression levels in the root under heat stress. In addition, *OsBES1s* exhibited upregulated levels in the shoot but no changes in the root under salt stress. It suggested that the expression of *OsBES1s* was probably tissue-specific or abiotic stress-response-specific. The mRNA of *GhBES1-4* and *-12* was highly expressed in roots, suggesting that they may also play key roles in cotton root development. The complete molecular mechanism and functions of these genes in root development required further exploration. The *GhBES1* family genes exhibited expression variations in response to one or more stress treatments. The transcript levels of *GhBES1-1*, *-4,* and *-12* were highly influenced by cold, hot, PEG, and salt stress compared to other *BES1* genes in cotton (Figure 4C). The overexpression of *GhBES1-4* in transgenic Arabidopsis and transient expression in *N. benthamiana* also showed a better growth state than WT under salt stress (Figure 6 and Appendix A). The *GhBES1-4* had a closer genetic relationship with *AtBES1-2* and *AtBES1-3*. Studies showed that *AtBES1-2* and *AtBES1-3* each had three homologous genes in Chinese cabbage [4,22], and their homologous genes were up-regulated under cold, heat, and PEG stress, with the highest expression level shown upon a treatment period of 12 h. Similarly enhanced expression patterns were observed in Arabidopsis, rice, and cotton, inferring their conserved roles in stress tolerance in different plant species. 

The protein interaction network reveals the regulatory relationships between proteins (Figure 7). Most of the proteins in the BES1 protein interaction network are related to BR metabolic pathways. For example, DWARF4 and CPD (CONSTITUTIVE PHOTOMORPHOGENESIS AND DWARFISM) are considered to encode rate-limiting enzymes in the BR biosynthesis pathway [23,24], activation of BSU1 phosphatase, dephosphorylation and inactivation of BIN2 kinase, and accumulation of nonphosphorylated transcription factor BZR1 in the nucleus. There are studies that have shown that DWF4 negatively regulates cold stress, and GSK1 and GSK2 can significantly improve the drought resistance of plants [25]. *AtMYB30*, encoding a R2R3-MYB TF, was known as a direct target of AtBES1 through microarray and chromatin immunoprecipitation experiments in Arabidopsis. The AtMYB30 protein binds to the promoter region of the BR target gene and unites with BES1 to adjust BR-prompted gene expression [26,27,28,29]. Under salt stress, the expression of the *GhBES1-4* gene was upregulated in transgenic plants, and so were its interaction proteins, such as BIN2, BIM1, BRI1, BKI1, and ELF6, demonstrating their potential functional role in salt stress resistance.

## 4. Materials and Methods

### 4.1. Sequence Retrieval and Gene Identification

The full-length sequences of *AtBES1* proteins were used as a query. The genomic and protein sequences of 27 plants, representative of nine most important plant ancestries, were obtained from freely available databases Phytozome v12.0 (https://phytozome.jgi.doe.gov/, accessed on 2 March 2022) and Congenie (http://congenie.org/, accessed on 2 March 2022) (for the gymnosperm *Picea sitchensis*). They were downloaded as a local protein database in order to identify BES1 homologs. In addition, the BES1-specific domain (PF05687) was used in a BLAST search to the native protein databases. Then, an HMM search (Biosequence analysis using profile hidden Markov Models, https://www.ebi.ac.uk/Tools/hmmer/search/hmmsearch, accessed on 2 March 2022) was conducted with the default parameter and an *e*-value ≤1 × 10^−5^. 

### 4.2. Bioinformatics and Protein Interaction Analysis

The phylogenetic trees were constructed from the BES1 proteins that had been retrieved using MEGAX (Molecular Evolutionary Genetics Analysis Version X, http://www.megasoftware.net, accessed on 2 March 2022). The sequence logos were analyzed by the online website MEME (Multiple Em for Motif Elicitation, https://meme-suite.org/meme/tools/meme, accessed on 2 March 2022). The gene structure of all *BES1* family genes was analyzed by the GSDS (Gene Structure Display Server 2.0) [30]. Collinearity analysis was performed by TBtools v1.098696 (https://github.com/CJ-Chen/TBtools, accessed on 2 March 2022). A DNA sequence of the length of 2000 bp upstream of the *BES1* gene in Arabidopsis, rice, and cotton was extracted by TBtools, and analyzed on PlantCARE (Plant *Cis*-Acting Regulatory Element, http://bioinformatics.psb.ugent.be/webtools/plantcare/html/, accessed on 2 March 2022), then visualized by TBtools. STRING (Version 11.0) [31] was used to construct a protein interaction network of Arabidopsis and the rice BES1 protein. The physicochemical characteristics of BES1 proteins were analyzed by Protscale, SignalP-5.0, and Phyre2 (Protein Homology/analogy Recognition Engine V 2.0).

### 4.3. Expression Data, Plant Materials, and Abiotic Stress Treatment

The expression profiles of *AtBES1* and *OsBES1* family genes in different developmental stages and tissues were determined, according to the transcribed records in Genevestigator and the Arabidopsis and rice eFP browsers in the Bio-Analytic Resource (http://bar.utoronto.ca/, accessed on 2 March 2022) database [32]. Expression data in nine tissues (root, stem, leaf, torus, stamen, pistil, calycle, fiber, and ovule), as well as under different abiotic stresses (PEG, salt, high, and low temperature), for *GhBES1* genes were obtained from transcriptome data. RNA-seq data were obtained from NCBI Sequence Read Archive (SRA: PRJNA248163). Based on iTRAQ data, proteomic changes of GhBES1 in cotton leaves were analyzed under salt stress. The data were deposited into CNGB Sequence Archive (CNSA) of the China National GeneBank DataBase (CNGBdb) with accession number CNP0002089 (https://db.cngb.org/, accessed on 2 March 2022).

The seedlings of Arabidopsis wild type (Col-0) were grown in a growth room under optimum growth conditions (16 h light and 8 h dark) at 20 °C for 30 days. The seedlings of japonica rice cultivar (*Oryza sativa* Nipponbare) were germinated at 28 °C for 72 h prior to moving to a growth room under 16 h light and 8 h dark at 28 °C for 16 days. The seedlings of upland cotton cv TM-1 were grown in an incubator under a 14 h light/10 h dark cycle at 28 °C until the three-leaf stage. All the plant seedlings were exposed to 4 °C for a period of 12 h as cold treatment. For heat treatment, Arabidopsis seedlings were exposed to 30 °C for a period of 12 h, whereas rice and cotton were grown at 40 °C for 12 h. Furthermore, the plant roots were immersed in the 175 mM NaCl and 20% PEG6000 solution for 12 h for salt and osmotic stress treatment, respectively. After treatments, all the sample plants were rapidly sampled and frozen in liquid nitrogen. Samples were stored at −80 °C for RNA extraction.

RNA was excavated by using the RNA Miniprep Kit (Axygen^®^ Corning Inc., Tewksbury, MA, USA) following the manufacturer’s instructions. Quantitative real-time PCR (qRT-PCR) was performed by using SYBR^®^ Premix Ex Taq™ II (TaKaRa, Dalian, China) on an Eppendorf Master cycler^®^ eprealplex (Eppendorf, Hamburg, Germany) system in order to examine the *BES1* expression levels. *Atactin*, *Osactin,* and *Ghactin* were selected as the internal controls. Three biological replicates, each of which contained three mechanical replications, were used.

### 4.4. Functional Verification of Transgenic Plants

In order to obtain transgenic rice plants, the full-length ORF of OsBES1-3 or OsBES1-5 was ligated into the KpnI and SacI sites of the plant expression vector pUN1301 vector (BioVector, Beijing, China) under the control of the ubiquitin promoter [33]. The recombinant plasmids were electroporated into Agrobacterium tumefaciens strain EHA105 cells and then introduced into rice embryonic Calli. 

PrimerX was used to design primers to modify the cDNA of *GhBES1-4*, and the mutated cDNA sequence was incorporated into the pBI121 vector, behind the CaMv 35S promoter. The constructed plasmid containing the mutated *GhBES1-4* was introduced into the *A. tumefaciens* strain (GV3101), which was used to generate transgenic plants [34]. The seeds of the T_3_ Arabidopsis plants expressing *GhBES1-4*, together with those of the wild-type control, were germinated in petri dishes containing 1/2 MS medium supplemented with 175 mM NaCl and vernalized at 4 °C for 48 h, prior to maintenance at 28 °C.

To elucidate the biological function of *GhBES1-4*, it was transiently expressed in *N. benthamiana* using the tobravirus pea early browning virus (PEBV)-based pCAPE expression system, together with associate and control plasmid pCAPE1 and pCAPE2. The vector constructions and *A. tumefaciens* transient transformation were performed as previously described [35]. After about three weeks of PEBV inoculation, samples of young *N. benthamiana* leaves were collected and floated on a 250 mM NaCl solution or water (control). The chlorophyll level of the *N. benthamiana* leaves was analyzed ten days after NaCl stress treatments, as described by Shabala [36].

## 5. Conclusions

In this study, we identified *BES1* in nine different plant lineages and performed functional characterization. The current *BES1* gene family, expanded mostly in angiosperm, seems to be a common ancestor of land plants. The spatial expression pattern analysis of the *BES1* genes, together with the expression patterns under different abiotic stresses, provided a basic resource for the examination of the molecular regulation of plant development and stress tolerance. As the conservation of biological functionality might be of ancestral origin, the functional conservation of the *BES1* gene family among different land plant species warrants a broad investigation in the future.

## Figures and Tables

**Figure 1 ijms-23-04242-f001:**
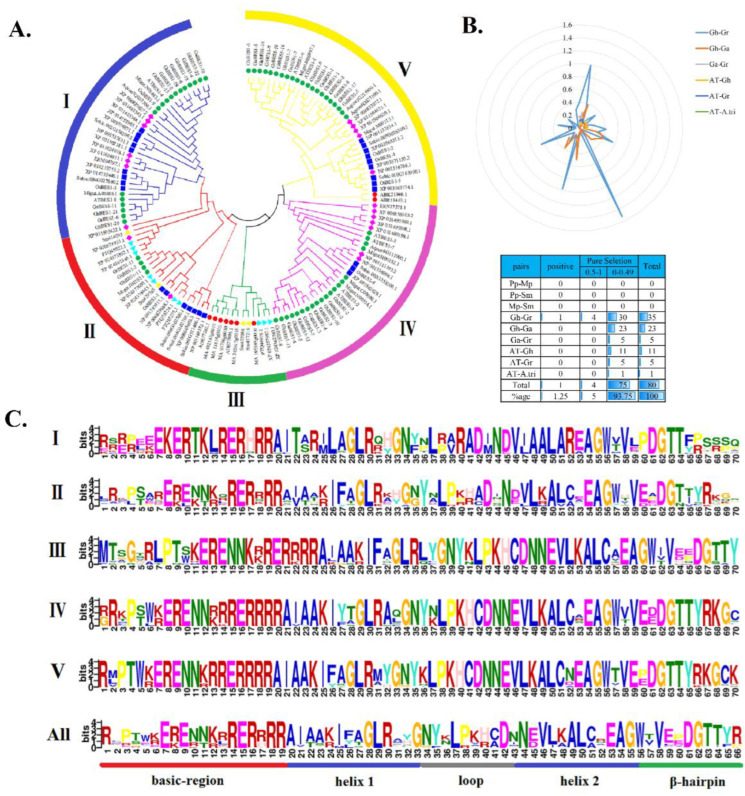
Phylogenetic tree, Ka/Ks values, and sequence features of *BES1*. (**A**) Phylogenetic tree of 135 *BES1* genes from 17 species. The ML method was adopted to construct the trees. Bootstrapping was performed 1000 times to gain support values for each branch. (**B**) Prediction of duplicated gene pairs involved in different combinations from the table shows the Ka/Ks ratio statistics of different genes. (**C**) The weblogo represents motifs of *BES1* protein sequences.

**Figure 2 ijms-23-04242-f002:**
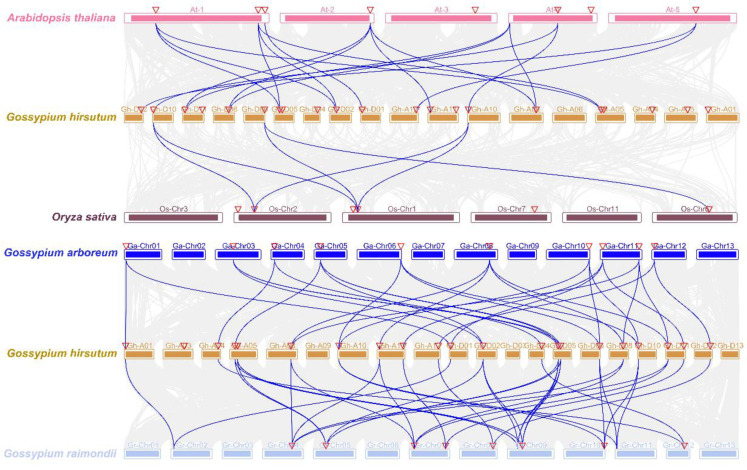
Collinearity analysis of *BES1* gene in Arabidopsis, rice, and cotton and among three cotton varieties (*G. arboretum*, *G. raimondii*, and *G. hirsutum*). The triangle represents the location of the *BES1* genes on the chromosome.

**Figure 3 ijms-23-04242-f003:**
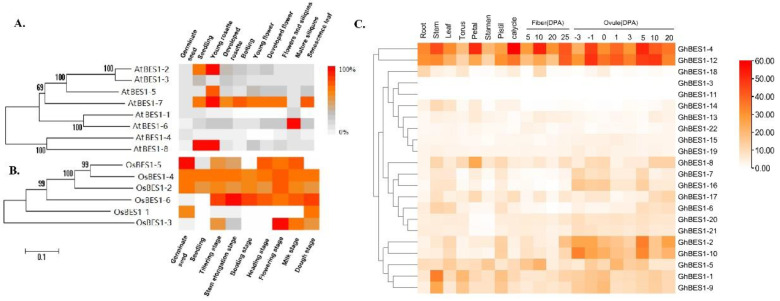
Heat map of *AtBES1*, *OsBES1,* and *GhBES1* family genes expression in different growing stages or tissues. (**A**) *Arabidopsis*; (**B**) rice; (**C**) *G. hirsutum*.

**Figure 4 ijms-23-04242-f004:**
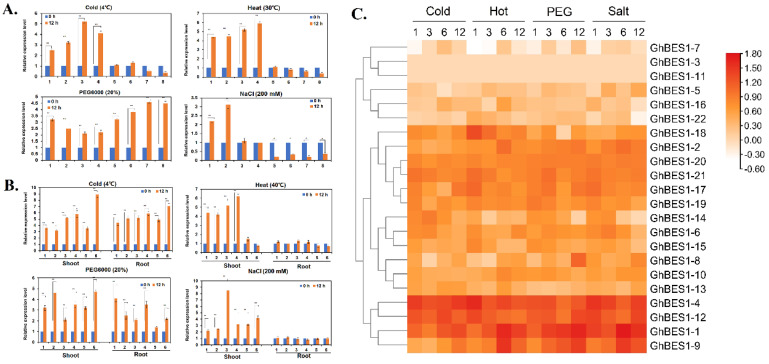
(**A**) Expression of *BES1* family genes in *Arabidopsis* under different stresses. Numbers on the X-axis represent *AtBES1s*, and 1~8 represent *AtBES1-1~AtBES1-8*. (**B**) Expression of *BES1* family genes in rice under different stresses. Numbers on the X-axis represent *OsBES1s*, and 1~6 represent *OsBES1-1~OsBES1-6*. (**C**) Expression of *GhBES1* family genes under different stress treatments. (*: 0.01 < *p* < 0.05, **: *p* < 0.01; the resulting mean values were presented as relative units. Error bar represents SD).

**Figure 5 ijms-23-04242-f005:**
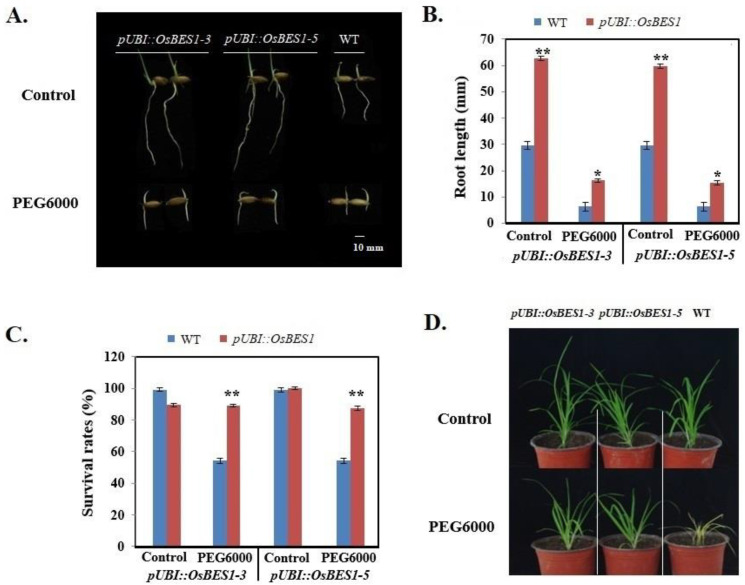
Functional verification in *OsBES1* genes under drought stresses. (**A**) The root lengths condition of WT, *pUBI::OsBES1-3,* and *pUBI::OsBES1-5* under control and drought stress. (**B**) Root length bar chart. ‘*’ indicates *p* < 0.05, ‘**’ indicates *p* < 0.01. Error bar represents SD. (**C**) The survival rate condition of WT, *pUBI::OsBES1-3,* and *pUBI::OsBES1-5* under control and drought stress. ‘**’ indicates *p* < 0.01. Error bar represents SD. (**D**) WT, *pUBI::OsBES1-3,* and *pUBI::OsBES1-5* under control and drought stress in pot.

**Figure 6 ijms-23-04242-f006:**
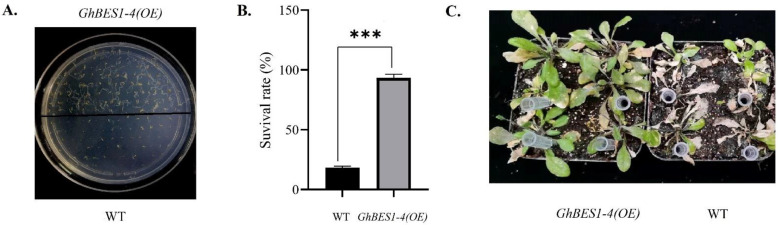
Functional verification of *GhBES1-4* gene under salt stress. (**A**) Growth of *GhBES1-4(OE)* (the upper part) and WT (the lower half) seeds on 175 mM NaCl plates. (**B**) The survival rate of WT and *GhBES1-4(OE)* under 175 mM NaCl. ‘***’ indicates *p* < 0.005. (**C**) WT and *GhBES1-4(OE)* under salt stress.

**Figure 7 ijms-23-04242-f007:**
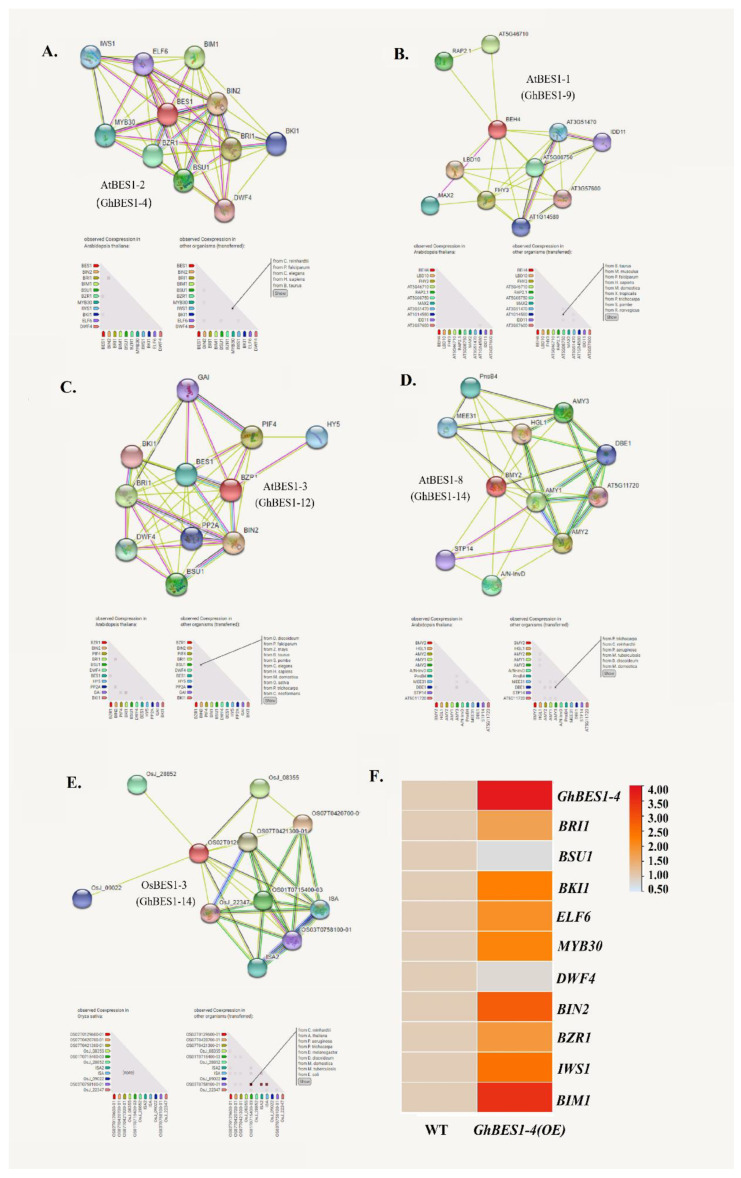
(**A**–**E**) Protein interaction of *AtBES1* and *OsBES1;* the two gene ids next to the central protein are Arabidopsis or rice BES1 ids and their homologous genes in *G. hirsutum.* (**F**) The expression pattern analysis of *AtBES1-2* interacting protein.

**Table 1 ijms-23-04242-t001:** Summary of the *BES1* transcription factors of representative plant species.

Lineage	Organism	Genome Size (Mb)	Gene Member	Number/Mb
Glaucophytes	*Cyanophora paradoxa*	0.14	0	0
Rhodophytes	*Galdieria sulphuraria*	13.7	0	0
	*Cyanidioschyzon merolae*	16	0	0
Chlorophytes	*Chlamydomonas reinhardtii*	121	0	0
	*Volvox carteri*	120	0	0
	*Coccomyxa subellipsoidea C-169*	48.8	0	0
	*Chlorella variabilis NC64A*	46.2	0	0
	*Micromonas pusilla RCC299*	21	0	0
	*Ostreococcus lucimarinus*	13.2	0	0
	*Ostreococcus tauri*	12	0	0
Bryophytes	*Physcomitrella patens*	511	6	0.0059
	*Marchantia polymorpha*	216.2	4	0.003
Lycophytes	*Selaginella moellendorffii*	100	4	0.0400
Gymnosperms	*Picea abies*	19,600	5	0.003
	*Picea sitchensis*	5500	4	0.002
Bacal angiosperm	*Amborella trichopoda*	699.2	9	0.005
	*Nymphaea colorata*	404.533	9	0.005
Monocots	*Zea mays*	2365	8	0.0034
	*Sorghum bicolor*	732.2	8	0.0109
	*Oryza sativa*	430	6	0.003
	*Brachypodium distachyum*	272	8	0.004
Eudicots	*Arabidopsis thaliana*	125	8	0.0640
	*Aquilegia coerulea*	306.5	4	0.0131
	*Gossypium raimondii*	775	11	0.0142
	*Mimulus guttatus*	430	8	0.0186
	*Gossypium hirsutum*	2430	22	0.0273
	*Gossypium arboretum*	752	11	0.0160

## Data Availability

Data is contained within the article or Appendix A.

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
