# Peer review of "Molecular Traits and Functional Exploration of BES1 Gene Family in Plants"

_ijms, 2022, doi:10.3390/ijms23084242_

Round 1

Reviewer 1 Report

The authors describe that genome comparison and structural analysis among several plant species including algae, expression patterns in various developmental stages as well as abiotic stress responses were investigated.

This Ms includes large volume of data and the results from experimental data the authors provided are clear and trustworthy. To date plant BES1 gene family is little known and I think this Ms has a big impact. However, to publish in IJMP adaptive revision is necessary by the reason described below.

L71   What is TINY? Full name of the abbreviation is needed at first mention in text.

Table1   Column of “Gene number” shows 0 in first nine organisms. It is difficult to determine zero gene copy number in genomes. What basis of zero gene number?

L166   “Cis” should be written in italics. Change other “cis-acting element” in text.  

Figure 4A and B   What mean these numbers of each number in X-axis?

Figure 4B   ã€€The #1 - #4 showed increased expression in shoot by heat stress but same or decreased expression levels in root. And also, by NaCl treatment #1 - #6 exhibited upregulated levels in shoot but nothing changes in root. The authors should explain these phenomenon.

Figure 5A, 5D and 6A-C   There is no measurable scale. The characters are hardly read in Fig 6A.

L252-258 and L266-271   Figure 6B must be 6C. The description about these figures is not match in text.

Figure 7   This figure has very small characters with low resolution. Therefore, I could not understand this figure.

L288   What are IWS, ELF and BIM? Full names are needed.

Discussion   It is hard to understand this part. That’s why the description without indicated Figures or Tables. The authors should devise overall in this part.

L298   What is meangiosperm? Is this typo?

L348   What is CPD?

L386   Plant is plant, not capital letter.

L400   About Oriza sativa Nipponbare, what kind of cultivar did authors used in this study?

Supplementary Figures and Tables   There is no Figure legend and Table explanation. The authors should add these.

Reviewer 2 Report

Yin et al. Reported on the characterization of the BRI1-EMSUSUPPRESSOR1 (BES1) gene family that act an important role during Brassinosteroids (BRs) signaling pathway. In particular, the authors studied the complex mechanisms of BES1-mediated developmental processes and abiotic stress tolerance.

Comments:

  • Why did the authors use for the salt tolerance screening 175 mM NaCl? Did they test also other concentrations? Or is it known in literature? Please clarify. Maybe the authors can add a salt tolerance analysis on MS medium with several concentrations of salt.

Figure 5A, scale bar is missing.

Figure 6, the authors showed WT and mutant seedlings on 175 mM NaCl plates but this image lacks of information. When WT and mutants under salt treatment are compared in pots, I suggest to add also the untreated.  Because from the image seems that that there is a drought stress and not a salt treatment. The control (untreated) should be also added in Fig 6A.

Figure 6A I would suggest to the authors to add an image with the seedling survival rate on 175 mM NaCl plates.

Figure 6B and C, there is an error in the caption or in the image. Check and modify for and adequate match between image and caption.  

Line 267, check the sentence with the image.

Figure 7. Impossible to read even with 500% zoom.

Round 2

Reviewer 2 Report

The authors have satisfactorily responded to all my questions and made the necessary changes to the manuscript.